# Drug-Resistant Stem Cells: Novel Approach for Colon Cancer Therapy

**DOI:** 10.3390/ijms23052519

**Published:** 2022-02-24

**Authors:** Nitin Telang

**Affiliations:** Cancer Prevention Research Program, Palindrome Liaisons Consultants, Montvale, NJ 07645-1559, USA; ntelang3@gmail.com

**Keywords:** colon cancer, drug-resistant stem cells, natural products

## Abstract

Background: Next to breast cancer, advanced stage metastatic colon cancer represents a major cause for mortality in women. Germline or somatic mutations in tumor suppressor genes or in DNA mismatch repair genes represent risk factors for genetic predisposition of colon cancer that are also detectable in sporadic colon cancer. Conventional chemotherapy for colon cancer includes combination of 5-fluoro-uracil with oxaliplatin and irinotecan or targeted therapy with non-steroid anti-inflammatory drugs and selective cyclooxygenase-2 inhibitors. Major limitations of these therapeutic interventions are associated with systemic toxicity, acquired tumor resistance and the emergence of drug resistant stem cells that favor initiation, progression and metastasis of therapy-resistant disease. These limitations emphasize an unmet need to identify tumor stem cell selective testable alternatives. Drug-resistant stem cell models facilitate the identification of new testable alternatives from natural phytochemicals and herbal formulations. The goal of this review is to provide an overview relevant to the current status of conventional/targeted therapy, the role of cancer stem cells and the status of testable alternatives for therapy-resistant colon cancer. Experimental models: Hyper-proliferative and tumorigenic cell lines from genetically predisposed colonic tissues of female mice represent experimental models. Chemotherapeutic agents select drug-resistant phenotypes that exhibit upregulated expressions of cellular and molecular stem cell markers. Mechanistically distinct natural phytochemicals effectively inhibit stem cell growth and downregulate the expressions of stem cell markers. Conclusions: The present review discusses the status of colon cancer therapy and inherent limitations, cancer stem cell biology, potential lead compounds and their advantages over chemotherapy. The present experimental approaches will facilitate the identification of pharmacological and naturally-occurring agents as lead compounds for stem cell targeted therapy of colon cancer.

## 1. Introduction

The progression of breast and colon cancer to advanced stage metastatic disease represent major causes of mortality in women. The American Cancer Society projects combined new female breast and colon cancer cases as 333,230 and cancer related deaths as 68,060 in 2022 [1]. Whereas it is well-established that estrogens represent a positive growth regulatory endocrine factor for breast cancer, the role of endocrine factors in colon cancer is less defined. It is however notable that the status of estrogen receptor-β (ER-β) functions as a prognostic factor for post-menopausal colon cancer and a target for colon cancer prevention [2,3]. Furthermore, in a preclinical model for genetically predisposed intestinal cancer the status of estrogen receptors ER-α and ER-β represents inhibitory modulators of colon cancer. The progeny generated from crossing of C57BL/6J-Min/+ with ER-α ^[+/−]^ and ER-β ^[+/−]^ mice represented ER-deficient Min/+ mice. These mice exhibited increased incidence in visible colon tumors and micro-adenomas. The expressions of ER-α and ER-β were regionally compartmentalized along the colonic crypt axis suggesting functional antagonism [4]. In addition, ER-β knockout mice exhibit enhanced incidence of pre-neoplastic colonic aberrant crypt foci in response to treatment with the chemical carcinogen, azoxymethane [5]. Collectively, this evidence provides important leads to identify the role of endocrine factors in female colon carcinogenesis.

Loss-of-function mutations in tumor suppressor genes and gain-of-function mutations in oncogenes represent ‘drivers’ of carcinogenic process. Germline mutation in the tumor suppressor adenomatous polyposis coli (APC) gene and in the DNA mismatch repair Mlh1, Msh2 and Msh6 genes represent risk factors for familial and hereditary colon cancer subtypes, respectively. A genetic defect in APC leads to chromosomal instability and aneuploidy [6], while that in DNA mismatch genes leads to microsatellite instability [7,8]. Somatic mutations in these genes are also detected in the sporadic colon cancer.

Conventional chemotherapy for colon cancer includes the use of the DNA anti-metabolite 5-fluoro-uracil (5-FU) in combination with platinum containing agents and topoisomerase inhibitors [9]. Targeted therapy for colon cancer uses non-steroidal anti-inflammatory drugs that target cyclooxygenases (COX-1 and COX-2), selective inhibitors of (COX-2) and selective inhibitors of ornithine decarboxylase (ODC), a rate-limiting enzyme in the polyamine biosynthesis pathway. The efficacy of these targeted therapeutic options has been documented in preclinical models. In the Apc ^Min^/+ model the non-steroidal anti-inflammatory drug sulindac inhibits intestinal polyp formation, induces cellular apoptosis, and inhibits COX-2 and prostaglandin E2 (PGE2) expressions [10]. In the same model administration of selective COX-2 inhibitor celecoxib (CLX) either early or late during the carcinogenic process results in the inhibition of polyp or adenoma formation, respectively, suggesting preventive or therapeutic efficacy [11]. These two agents have documented clinical efficacy in patients with genetically predisposed and sporadic colon cancer [12,13].

Major limitations with conventional and targeted therapies are associated with systemic toxicity, acquired tumor resistance and the emergence of drug-resistant cancer stem cells that have a negative impact on patient compliance, and facilitate the progression of therapy resistant diseases. These limitations emphasize the development of reliable drug-resistant cancer stem cell models and the identification of non-toxic cancer selective testable alternatives.

Dietary modulations with low fat, high fruits/vegetables and high micro-nutrient content have documented cancer protective effects. Naturally-occurring phytochemicals and nutritional herbs, functioning via distinct molecular mechanisms, have documented growth inhibitory efficacy in preclinical models for colon and breast cancers [14,15,16,17,18]. As a result of low toxicity, documented mechanistic leads for their efficacy, and documented human consumption, natural products may represent testable alternatives for therapy resistant breast and colon cancer [19,20,21,22].

The present review provides an overview of conventional/targeted chemotherapy for colon cancer, its limitations, and utility of natural phytochemicals and their advantages over conventional chemotherapy. Furthermore, this review discusses experimental evidence relevant to: (i) Development and characterization of cellular models for genetically predisposed colon cancer; (ii) Growth inhibitory efficacy of naturally-occurring and pharmacologic agents; (iii) Development and characterization of drug-resistant stem cell models; and (iv) Leads for efficacy of naturally-occurring agents on cancer stem cells.

## 2. Experimental Models

The mouse models for genetically predisposed intestinal neoplasia exhibit adenoma formation predominantly in the small intestine. However, in the clinical familial adenomatous polyposis (FAP) and hereditary non-polyposis colon cancer (HNPCC) syndromes, the adenoma/adenocarcinoma formation is detectable in the colon. Development and characterization of cellular models provide cells with clinically relevant genetic defects and a quantifiable risk of colon cancer. This approach represents novel experimental systems directly relevant for investigations on colon carcinogenesis and its prevention/therapy.

Colonic epithelial cell culture models are developed from the female wild type mice and those with genetic predisposition for intestinal cancer. Colonic mucosal epithelium from the descending colon of the wild type C57BL/6J mice was used to develop the C57 COL cell line [23,24]. The 850 ^MIN^/+ exhibits high incidence of multiple intestinal neoplasia due to a germ line mutation in the codon 850 of the tumor suppressor Apc gene [25]. This cell line represents a model for FAP [23,24]. The Apc 1638N ^[+/−]^ mouse exhibits high incidence of small intestinal adenoma due to a germ line mutation in the codon 1638 of the Apc gene [26]. The 1638N COL line represents a model for FAP [27]. The Mlh_1_ ^[+/−]^ mouse exhibits a germ line mutation in the Mlh_1_ DNA mismatch repair gene [28]. The Mlh_1_ COL cell line represents a model for HNPCC. The Mlh_1_ ^[+/−]^/1638 ^[+/−]^ mouse exhibits germline mutation in the Mlh1 and Apc 1638 genes [28]. This cell line represents a model for HNPCC [22]. The origin and genotypic and biological characteristics of the colonic epithelial cell lines are presented as illustrative examples in Table 1.

It is notable that the non-tumorigenic cell lines lack anchorage independent (AI) colony formation in vitro and tumor formation in vivo, while the tumorigenic cell lines exhibit the two properties.

The human colon cancer models HCT-116 and SW480 exhibit distinct genotypes that are relevant to clinical sporadic colon cancer. The HCT-116 cells exhibit wild type APC^WT^ and mutant β-catenin ^MT^ genotype. The SW480 cells exhibit APC ^MT^ and β-catenin ^WT^ genotype. Somatic mutations in APC and β-catenin genes are commonly observed in sporadic colon cancer. Thus, these human tissue derived cellular models provide valuable clinically relevant experimental approaches.

### 2.1. Mechanistic Assays

The optimized mechanistic assays include flow cytometry for cell cycle progression and cellular apoptosis, and quantitative immuno-fluorescence assay for Apc target proteins and molecular markers for drug-resistant stem cells [19,21,22,23,24].

### 2.2. Test Agents

The synthetic pharmacological agents, celecoxib (CLX), difluoro-methyl ornithine (DFMO), 5-fluoro-uracil 5-FU) and sulindac (SUL) have documented clinical and preclinical efficacy for colon cancer [9,12,13,29,30]. Naturally-occurring polyphenol, N-3 polyunsaturated fatty acids and curcumin have documented chemo-preventive efficacy in the preclinical model for FAP, and are documented to enhance the effects of chemotherapeutic agents [14,15,31,32]. The test agents used in the experiments with the 850 ^MIN^ COL model are listed in Table 2.

## 3. Carcinogenic Transformation

### 3.1. Growth Pattern and Molecular Markers in Colon Models

The comparative growth pattern of the non-tumorigenic C57 COL and the tumorigenic 850 ^MIN^ COL cells are illustrated in Table 3. This illustration indicates that the tumorigenic 850 ^MIN^ COL cells exhibit loss in homeostatic growth control and gain in cancer risk.

The molecular characteristics of tumorigenic 850 ^MIN^ COL cells are illustrated in Table 4. These cells exhibit loss of Apc and resultant upregulation of selected Apc target proteins. The data presented in Table 3 and Table 4 taken together, suggest that aberrant proliferation and increased cancer risk may, in part be due to upregulation of Apc target proteins.

### 3.2. Combinatorial Anti-Proliferative Effects on FAP Model

Mechanistically distinct multi-drug combinations represent a common treatment option for colon cancer chemotherapy [9]. In this option the constituent drugs synergize to enhance their individual efficacy. The Apc ^Min^/+ mouse represents a well-established preclinical model for FAP syndrome. In this model low dose combinations of mechanistically distinct chemo-preventive agents have documented enhanced efficacy. Thus, low dose combination of piroxicam (PIROX) and difluoro methyl ornithine (DFMO) results in inhibition of polyp formation compared to these agents administered individually [33]. Low dose combination of celecoxib and docosa hexaenioc acid (DHA) in human colonic adenoma HCA-7 cells demonstrate enhanced anti-proliferative and pro-apoptotic effects that are associated with reduction in COX-2 and prostaglandin E2 (PGE2) [34]. In the Apc ^Min^/+ model combination of atorvastatin (ATOR) with CLX inhibits polyp formation and increases the rate of apoptosis. These effects are associated with reduction in the molecular targets of these agents [35]. DFMO functions as a selective inhibitor of ornithine decarboxylase (ODC), an established early response gene, a transcriptional target of c-Myc and a rate limiting enzyme for polyamine biosynthesis [29]. CLX functions as a selective inhibitor of COX-2, an Apc target gene [11,13], SUL functions as a pan inhibitor of COX [12], epigallocatechin gallate (EGCG) functions as a modulator of epidermal growth factor receptor (EGFR) signaling [15,32], and eicosa pentaenoic acid (EPA) and DHA function as inhibitors of lipoxygenase, a critical enzyme for leukotrienes [34]. Thus, loss of function mutation in tumor suppressor Apc may in part, be responsible for disruption of normal Wnt/β-catenin signaling pathway in the 850 ^MIN^/+ model. Reversal of the abnormal Wnt/β-catenin pathway by chemo-preventive agents may be responsible for mechanistic efficacy of these agents.

The experiment illustrated in Figure 1 examined the inhibitory effects of DFMO in combination with EGCG, EPA, SUL and CLX. Low concentrations of these agents used as single agents resulted in a minimal inhibition of AI colonies. In contrast, the low dose combinations resulted in a substantially greater inhibition of AI colonies.

The mainstream treatment options of conventional and targeted therapy for both breast and colon cancer are associated with systemic toxicity and acquired tumor resistance leading to progression of therapy-resistant disease. These limitations emphasize identification of nontoxic testable alternatives against therapy-resistant disease.

Herbal formulations consisting of multiple Chinese nutritional herbs are widely used in traditional Chinese medicine for treatment of triple negative breast cancer (TNBC) and their growth inhibitory efficacy is predominantly via inhibition of Phospho-inositidyl-3 kinase (PI3K), protein kinase B (AKT), molecular target of rapamycin (mTOR), mitogen activated protein kinase (MAPK) and Wnt/β-catenin pathways [36]. In the MDA-MB-231 model for TNBC anti-proliferative and pro-apoptotic effects of several mechanistically distinct Chinese nutritional herbs are due to inhibition of retinoblastoma (RB), cyclin dependent kinase (CDK), extra cellular signal-regulated kinase (ERK), PI3K and AKT pathways and induction of pro-apoptotic caspase activity [17,18,37].

Extracts from roots, fruits, leaves, and seeds of nutritional herbs have documented in vitro and in vivo growth inhibitory effects on several colon cancer models via multiple mechanisms [38]. The herbal preparation BP3B composed of three herbal extracts exhibits anti-proliferative, pro-apoptotic and anti-angiogenic effects in a patient-derived colon tumor xenograft model [39]. The anti-tumor effect of the herbal formulation BP10A is associated with enhanced therapeutic efficacy of irinotecan and oxaliplatin in a patient-derived colon tumor xenograft model [40]. Herbal medicines containing plant polyphenols exhibit in vitro anti-proliferative and pro-apoptotic effects on colon carcinoma-derived HCT116, HCT 15 and CT26 cell lines, however, not on the normal colon cell line NCM460. The growth inhibitory effects involved receptor tyrosine kinase, MAPK-ERK (MEK) and nuclear factor kB (NFkB) pathways [41]. Notably, the nutritional herbs that are used in the traditional Chinese medicine for gastro-intestinal diseases also function as natural estrogenic agents and anti-inflammatory agents [38].

In the present 850 ^MIN^ COL model, select Chinese nutritional herbs exhibit concentration-dependent anti-proliferative effects, and at their respective maximum cytostatic inhibitory concentration IC_90_, inhibit AI colony formation (unpublished observations). These preliminary observations are currently being extended to identify mechanistic leads that may involve specific molecular pathways and targets responsible growth inhibitory efficacy.

### 3.3. Characterization of Drug-Resistant Stem Cell Model

Epithelial stem cells represent an essential component of the proliferative zone in all epithelial organ sites responsible for normal cellular homeostasis [42]. In contrast, cancer stem cells represent an essential component of therapy-resistant cancers [43].

In stem cells of normal colonic crypts Wnt/β-catenin/Apc signaling pathway is critical for the regulation of cell proliferation/cyto-differentiation/apoptosis during regeneration and cellular homeostasis of mucosal epithelium. During the progression of normal epithelium to adenoma/adenocarcinoma the stem cells exhibit a loss in function mutations in the Apc gene, upregulated expression of Apc target genes and disruption of tumor suppressive signaling cascade. Thus, with colon carcinogenesis the Wnt/β-catenin signaling pathway in general, and the loss-of-function mutation in APC gene in particular, play an important role. In the Apc signaling pathway status of APC target genes including cyclin D1, c-Myc, cluster of differentiation 44 (CD44) and COX-2 represent relevant quantitative end points not only for colon carcinogenesis, but also for effective response to therapy.

Several assays for isolation of putative stem cells have been optimized. Commonly utilized assays include: (i) Drug efflux positive side population; (ii) Alcohol dehydrogenase-1 (ALDH-1) positive cell population; (iii) Tumor spheroid formation; and (iv) Drug-resistant phenotypes. The frequency of drug resistant phenotypes leading to acquired resistance is commonly observed in advanced stage metastatic clinical colon cancer [9,30]. Optimized assays to isolate and characterize drug-resistant stem cells have been published [19,22,44,45].

A combination of in vitro tissue culture and in vivo transplantation approaches have been used to isolate and characterize colon cancer stem cells. Drug resistant cells exhibit increased incidence in stem cell specific tumor spheroid formation, upregulated expressions of cell surface markers CD44 and CD133, and nuclear transcription factor c-Myc in vitro [19,21,22]. Tumor spheroid models derived from sporadic colon cancer cell lines have been utilized to identify stem cell specific molecular markers including octamer binding transcription factor-4 (Oct-4), Kruppel-like factor-4 (Klf-4), sex determining region Y-box-2 (SOX-2) and cellular Myc (c-Myc) [44,45]. Patient derived colon cancer stem cell-enriched spheroids have been utilized to identify potential therapeutic targets [39,40]. At the biological level, cancer stem cell derived tumor spheroid formation in vitro [44,45], and stem cell mediated cancer initiation in vivo has been documented [46,47,48].

Long-term treatment with chemotherapeutic agents has been used to select putative drug-resistant stem cells. These stem cells are characterized by quantifying the status of select stem cell markers. Figure 2 illustrates stem cell marker expression in sulindac sensitive (SUL-S) and sulindac resistant (SUL-R) cells isolated from the parental 850 ^MIN^ COL cells. Stem cell markers tumor spheroids (TS), CD44, CD133 and c-Myc represent established cellular and molecular markers for stem cells [19,21,22]. Upregulated expressions of these stem cell specific markers in the SUL-R phenotype suggest an acquisition of stem cell properties.

Figure 3 illustrates stem cell marker expressions in 5-fluoro-uracil sensitive (5-FU-S) and 5-fluoro-uracil resistant (5-FU-R) cells isolated from the parental Mlh_1_/1638N COL cells. Upregulated expressions of the stem cell markers in the 5-FU-R phenotype suggest acquisition of stem cell properties.

### 3.4. Tumor Spheroid Inhibition by Dietary Agents

Several Chinese herbal medicines and their constitutive active components, as well as naturally occurring phytochemicals including polyphenols, flavonoids and terpenoids, have been documented to exhibit stem cell selective growth inhibitory effects in models for pancreas, prostate and breast cancers. The mechanism of action of these agents have been documented to involve Wnt/β-catenin, sonic hedgehog, Notch, PI3K/AKT/mTOR and NFkB survival pathways that are activated in cancer cells. In addition some of these phytochemicals effectively augment the efficacy of conventional chemotherapeutic drugs [49,50,51,52,53].

Tumor spheroid formation represents a specific cellular marker for stem cells. The inhibitory effects of select naturally-occurring dietary agents on tumor spheroid formation in the 850 ^MIN^ COL SUL-R cells are illustrated in Table 5. Treatment with these agents reduced the tumor spheroid number. These data provide mechanistic leads for the effects of natural compounds on drug-resistant stem cell population.

### 3.5. Effect of Curcumin on Stem Cells

In stem cell models for the prostate and pancreas, natural products such as sulforaphane and quercetin have been documented to augment the therapeutic efficacy of cisplatin, gemcitabine, doxorubicin, 5-fluoro-uracil and the multi-kinase inhibitor sorafinib. In these studies, stem cell markers such as tumor spheroids, ALDH-1, Notch-1 and c-REL represent the quantitative endpoint parameters [54,55,56].

Curcumin (CUR) is a bioactive agent present in the root of the turmeric plant, and its anti-proliferative effects are due to multiple mechanisms that include inhibition of Wnt/β-catenin signaling, NFkB signaling, PI3K/AKT/mTOR signaling and COX-2 activity [52,55]. The effect of CUR on the SUL-R stem cells from the 850 ^MIN^ COL model is illustrated in Figure 4. The inhibition of select stem cell markers provides potential mechanistic leads for the efficacy of efficacious natural products.

Consistent with this evidence it is notable that several mechanistically distinct naturally-occurring compounds have documented stem cell specific inhibitory effects [51,52,53,54,55].

## 4. Conclusions

The use of conventional and targeted therapy for colon cancer is limited by systemic toxicity, acquired tumor resistance and the emergence of drug-resistant stem cells. Dietary natural products and herbal formulations, due to their lack of systemic toxicity and documented human consumption, may represent testable alternatives for therapy resistant colon cancer. Patients presenting with advanced stage therapy-resistant breast and colon cancer frequently seek to follow complementary and alternative medicine. Herbal formulations from folk medicines used by the indigenous native population in the regions of Latin America, India and China are common [36,38]. However, the evidence for their efficacy is predominantly anecdotal, and lacks relevant peer reviewed clinical evidence. In contrast, herbal formulations common in traditional Chinese medicine are most frequently used in Europe and other regions. The cellular models are developed from genetically predisposed colonic epithelium [23,24,27], stem cell models are developed from drug-resistant colonic epithelial cells [19,21,22], and tumor spheroid models are developed either from patient derived cancer stem cell-enriched spheroids [39,40] or from sporadic cancer derived cell lines [44,45]. Experimental modulations are quantified by specific and sensitive end point markers, such as cell membrane proteins and nuclear transcription factors. These aspects validate novel experimental approaches to identify new stem cell targeting pharmacological agents and potential natural products as lead compounds. These agents may represent efficacious alternatives for stem cell targeted therapy of colon cancer. The present experimental approaches may identify clinically translatable mechanistic leads applicable to patients.

An overview of conceptual background and experimental evidence discussed in the present review is summarized in Figure 5.

## 5. Future Prospects

The experimental evidence discussed in the present review provides a rationale for future research directions involving ex vivo investigations using therapy resistant patient derived tumor spheroid models [39,40], and patient derived organoid models [57,58,59,60]. In addition, recent investigations on cancer stem cells are focused on examining the molecular mechanisms responsible for Wnt signaling in Apc and β-catenin mutated colorectal cancer cell organoids [61,62,63,64,65], the role of DNA repair in stem cell renewal [66], mechanisms for regulation of CD44 in stem cell proliferation [67], the role of one-carbon metabolism in cancer stem cells [68], and the role of ER-β in mammary carcinogenesis [69]. The Wnt/β-catenin signaling pathway plays important roles in colon cancer metastasis, cancer stem cell biology and epithelial-mesenchymal transition process, and therefore, a valuable therapeutic target [70,71]. Retinoic acid receptor agonists may function as novel inhibitors of colon cancer progression [72]. ER-β agonists may function as novel ER selective agents [73].

Published evidence, discussed above, provides a scientifically robust rationale for future research directions to identify new cancer therapeutic agents and discover clinically translatable mechanistic links. Preclinical investigations using patient tumor tissue derived xenograft (PDTX) models and patient tumor tissue derived organoid (PDTO) models may establish relevance for clinical secondary prevention/therapy for colon cancer patients.

## Figures and Tables

**Figure 1 ijms-23-02519-f001:**
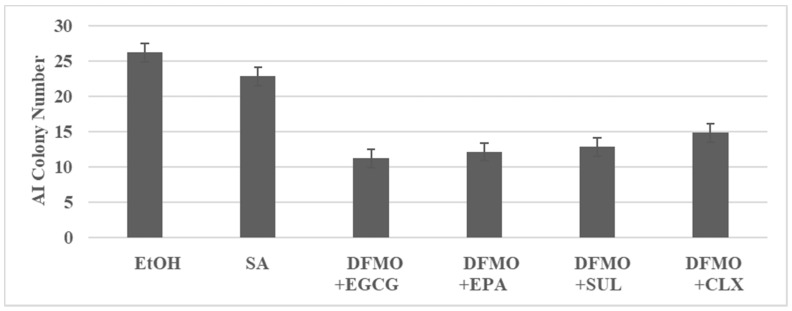
Combinatorial effects of test agents in 850 ^MIN^ COL model. AI colony number is determined at day 14 after seeding 100 cells. Data presented as mean ± SD, n = 12 per treatment group, and analyzed by ANOVA with Dunnett’s multiple comparison test (α = 0.05). EtOH, ethanol; SA, single agent; DFMO, difluoromethyl ornithine; EGCG, epigallocatechin gallate; EPA, eicosapentaenoic acid; SUL, sulindac; CLX, celecoxib; ANOVA, analysis of variance. Data summarized from [24].

**Figure 2 ijms-23-02519-f002:**
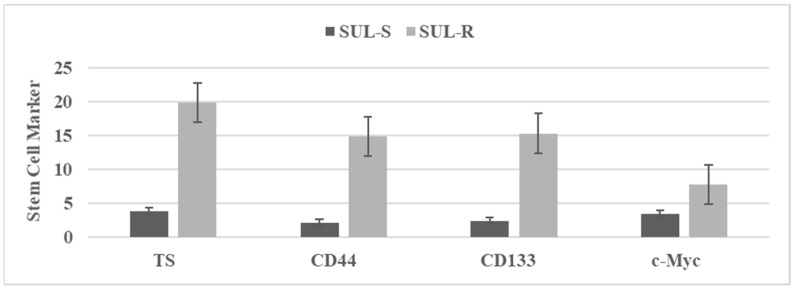
The status of stem cell markers in sulindac resistant (SUL-R) 850 ^MIN^ COL phenotype. Figure 2. TS: Tumor spheroid number determined at day 14 after seeding 100 cells. Mean ± SD, n = 3 per treatment group. CD44, CD133, c-Myc. RFU determined at day four after seeding 1.0 × 10^5^ cells. Data presented as mean ± SD, n = 3 per treatment group, and analyzed by paired Student’s *t*-test. SUL-S, sulindac sensitive; SUL-R, sulindac resistant; CD, cluster of differentiation; c-Myc, cellular Myc; RFU, relative fluorescent unit.

**Figure 3 ijms-23-02519-f003:**
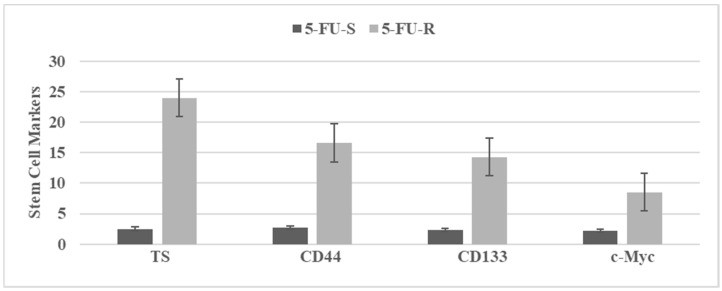
The status of stem cell markers in 5-fluoro-uracil resistant (5-FU-R) Mlh_1_/1638N COL phenotype. Figure 3. TS: tumor spheroid number determined at day 14 after seeding 100 cells. Data expressed as Mean ± SD, n = 3 per treatment group. CD44, CD133 c-Myc. RFU determined at day four after seeding 1.0 × 10^5^ cells. Data expressed as Mean ± SD, n = 3 per treatment group. Data analyzed by paired Student’s *t*-test. 5-FU-S, 5-fluoro-uracil sensitive; 5-FU-R, 5-fluoro-uracil resistant; CD, cluster of differentiation; c-Myc, cellular Myc; RFU, relative fluorescent unit. Data summarized from [22].

**Figure 4 ijms-23-02519-f004:**
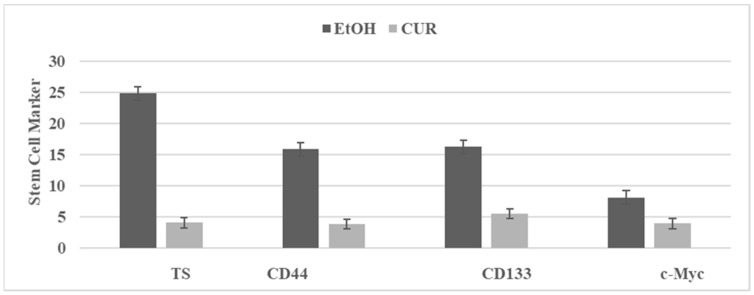
Effect of curcumin on stem cell markers in SU-R 850 ^MIN^ COL phenotype. TS: Tumor spheroid number determined at day 14 after seeding of 100 cells. Mean ± SD, n = 3 per treatment group. CD44, CD133, c-Myc. RFU determined at day four after seeding 1.0 × 10^5^ cells. EtOH, ethanol; CUR, curcumin; CD, cluster of differentiation; c-Myc, cellular Myc; RFU, relative fluorescent unit.

**Figure 5 ijms-23-02519-f005:**
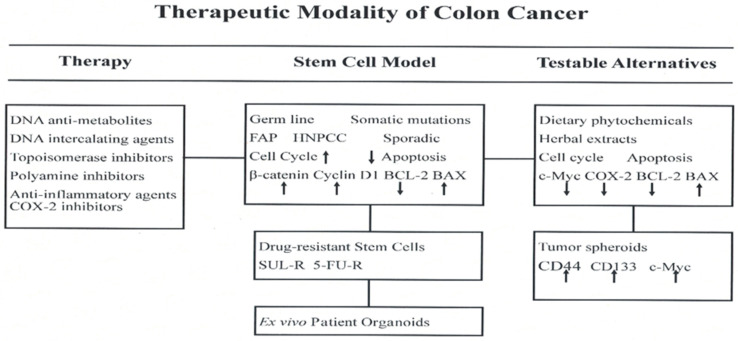
COX-2, cyclo-oxygenase-2; FAP, familial adenomatous polyposis; HNPCC, hereditary non-polyposis colon cancer; c-Myc, cellular myc; SUL-R, sulindac resistant; 5-FU-R, 5-fluoro-uracil resistant; BCL-2, B cell lymphoma-2; BAX, BCL-2 associated X protein; CD, cluster of differentiation.

**Table 1 ijms-23-02519-t001:** Cellular models for colonic epithelial cells.

Cell Line	Genotype		Origin ^a^	Biological Characteristics	Model	Reference
	Mlh1	Apc				
C57 COL	+/+	+/+	C57BL/6J	Immort. NT	Normal colon	[23]
850 ^MIN^ COL	+/+	+/−	Apc ^Min^/+	Immort. T	FAP	[23,24]
1638N COL	+/+	+/−	1638N ^[+/−]^	Immort. T	FAP	[27]
Mlh1 COL	+/−	+/+	Mlh1 ^[+/−]^	Immort. NT	HNPCC	[28]
Mlh1/1638N COL	+/−	+/−	Mlh1 ^[+/−]^ 1638N ^[+/−]^	Immort. T	HNPCC	[22]

^a^ developed from the colonic mucosal epithelium of female mice. Mlh_1_, DNA mismatch repair gene; Apc, adenomatous polyposis coli; Immort, spontaneous immortalization; NT, non-tumorigenic; T, tumorigenic; FAP, familial adenomatous polyposis syndrome; HNPCC, hereditary non-polyposis colon cancer.

**Table 2 ijms-23-02519-t002:** Test Agents.

Agent	Source/Origin	References
Carnosic Acid (CA)	Rosemary terpenoid	[20]
Celecoxib (CLX)	Synthetic COX-2 inhibitor	[11]
Curcumin (CUR)	Curcuma longa root	[14]
Difluoromethyl ornithine (DFMO)	Synthetic ODC inhibitor	[29]
Docosa hexaenoic Acid (DHA)	Fish oil N-3 PUFA	[33]
Eicosa pentaenoic Acid (EPA)	Fish oil N-3 PUFA	[34]
Epigallo catechin gallate (EGCG)	Green tea polyphenol	[15,32]
Sulindac (SUL)	Synthetic COX-1/COX-2 inhibitor	[10]

COX, cyclooxygenase; ODC, ornithine decarboxylase; N-3 PUFA, omega-3 polyunsaturated fatty acid.

**Table 3 ijms-23-02519-t003:** Hyper-proliferation and carcinogenesis in 850 ^MIN^ COL model.

Marker	Cellular Models			
	C57 COL	850 ^MIN^ COL	*p* Value	Relative to C57 COL
PDT (h.) ^a^	34.0 ± 3.8	14.0 ± 1.6	0.032	−58.9%
Sat. Den. (×10^5^) ^b^	7.7 ± 0.5	67.9 ± 7.5	0.001	+7.8x
G1: S + G2/M ^c^	3.1 ± 0.3	0.6 ± 0.2	0.010	−80.0%
Aneuploidy (%) ^c^	Not detected	81.7 ± 9.1		
Sub G0 (%) ^c^	4.3 ± 0.1	0.8 ± 0.3	0.010	−81.4%
AI Colonies ^d^				
Incidence	0/18	18/18		
Number	----	18.9 ± 2.5		
Tumor				
Incidence	0/10	8/10		
Latency	24 weeks	3–5 weeks		

^a^ determined from exponential growth phase. ^b^ determined at day seven after seeding 1.0 × 10^5^ cells. ^c^ determined at day four after seeding 1 × 10^5^ cells by cell cycle analysis. ^d^ determined at day 14 after seeding 100 cells. Data presented as mean ± SD, n = 3 per treatment group, and analyzed by Student’s *t*-test. PDT, population doubling time; Sat. Den., saturation density; AI, anchorage independent; SD, standard deviation. Data are summarized from [21].

**Table 4 ijms-23-02519-t004:** Molecular markers in C57 COL and 850 ^MIN^ COL models.

Marker ^a^	Cellular Model			
	C57 COL	850 ^MIN^ COL	*p* Value	Relative to C57 COL
Apc	17.6 ± 1.7	Not detected		
β-catenin	4.7 ± 0.3	9.0 ± 0.8	0.001	+91.5%
Cyclin D1	3.0 ± 0.1	16.1 ± 1.5	0.001	+4.4x
c-Myc	2.1 ± 0.2	6.6 ± 0.4	0.010	+2.1x
COX-2	3.9 ± 0.4	12.8 ± 1.8	0.010	+2.3x

^a^ determined at day four after seeding 1.0 × 10^5^ cells. Data presented as RFU mean ± SD, n = 3 per treatment group by quantitative immune-fluorescence assay, and analyzed by Student’s *t*-test. COX-2, cyclooxygenase-2; SD, standard deviation. RFU; relative fluorescent unit. Data are summarized from [21].

**Table 5 ijms-23-02519-t005:** Effects of dietary agents on tumor spheroid formation in 850 ^MIN^ COL model.

Dietary Agent	Tumor Spheroid Number ^a^	*p* Value	Relative to Solvent Control
EtOH	24.8 ± 2.2		
CUR	4.0 ± 0.4	0.010	−83.9%
EGCG	6.0 ± 1.3	0.030	−68.8%
EPA	6.4 ± 1.4	0.030	−66.7%
DHA	6.8 ± 1.4	0.030	−64.5%
CA	8.6 ± 1.9	0.042	−55.5%

^a^ determined at day 14 after seeding 100 cells. Data expressed as mean ± SD, n = 3 per treatment group and analyzed by one-way ANOVA with Dunnett’s multiple comparison test (α = 0.05). SD, standard deviation; ANOVA, analysis of variance. EtOH, ethanol; CUR, curcumin; EGCG, epigallocatechin gallate; EPA, eicosa pentaenoic acid; DHA, docosa hexaenoic acid; CA, carnosic acid.

## Data Availability

The data sets used in the present review are available from the author on reasonable request.

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
