# Peer review of "Drug-Resistant Stem Cells: Novel Approach for Colon Cancer Therapy"

_ijms, 2022, doi:10.3390/ijms23052519_

Round 1

Reviewer 1 Report

This review aimed to comprehensively summarize the status of colon cancer therapy and inherent limitations, cancer stem cell biology, potential lead compounds and their advantages over chemotherapy. The manuscript was well prepared and the conclusion was sound. It will provide scientifically rationale for future researches to find new cancer therapeutic agents and clinically translatable approaches. Therefore, it will be recommended to the journal to help the scientific community to develop new drugs and methodologies for colon cancer. However, a minor revision should be made before publication.

Some suggestions for promotion:

  1. The herbal medicines from Europe and other regions that related to the colon cancer therapies should also be discussed.
  2. Page 2 line 72, “disease” should be “diseases”.
  3. Page 2 line 78, correct “cancer” to “cancers”.
  4. Page 5 line 178, the double dot should be changed to a single dot.
  5. Page 6 lines 225-229, what is the meaning of bold for these sentences.
  6. As for the figures, their original literatures should be cited and indicated among the legends under illustrations.
  7. The abbreviation of references should be checked, such as the journal of “Cancer Letts” in ref 2 should be “Cancer Lett.”

Author Response

  1. Comment: The herbal medicines from Europe and other regions related to colon cancer therapies should be discussed.

Response: Patients presenting with advanced stage therapy-resistant breast and colon cancer frequently seek to follow complementary and alternative medicine. This alternative includes the use of natural products, nutraceuticals, herbal medicines and herbal formulations. Herbal formulations from folk medicines used by the indigenous native population in regions of Latin America, India and China are common. However, the evidence for their efficacy is predominantly anecdotal and lacks adequate peer reviewed clinical evidence. In contrast, herbal formulations from traditional Chinese medicine are most frequently used in Europe and other regions. This aspect is included in the text of revised version of the manuscript. 

  1. Comment: Page 2, line 72: “disease” should be corrected to “diseases”.

Response: This corrections is made.

  1. Comment: Page 2, line 78: “cancer” should be corrected to “cancers”.

Response: This correction is made.   

  1. Comment: Page 5, line 178: the double dot should be changed to single dot.

Response: This correction is made.

  1. Comment: Page 6, Lines 225-229: The sentences in bold face are confusing.

Response: This segment in was included in bold face because it was a specific response to the reviewer comment. This aspect is now corrected.

  1. Comment: Original literatures should be cited in the figure legends.

Response: The origin of the summarized primary data is included as relevant reference numbers in the legends. The unpublished primary data is clearly indicated as such in the legends.

  1. Comment: Journal abbreviations in the references should be checked.

Response: The abbreviations are checked and are corrected.

Reviewer 2 Report

The manuscript is difficult to follow. The goals should be clearly stated. The abstract should have a background, methods, results and a conclusion. 

Author Response

  1. Comment: The manuscript is difficult to follow. The goals should be clearly stated.

Response: The goal for the present review article is to provide an overview relevant to the current status of conventional/targeted therapy for colon cancer, role of cancer stem cells and status of testable alternatives for therapy resistant colon cancer. This is now clearly started in the abstract and introduction sections.

  1. Comment: The abstract should have a background, methods, results and a conclusion.

Response: This recommendation is more relevant to a research article. Traditionally in the review article specific sections for methods and results are not included. However, pursuant to the recommendation, original abstract is now revised to a structured abstract that contains the Background, Experimental models and Conclusions sections.